# Risk of Ulnar Nerve Injury Following Caudo-Medial Arthroscopic Portal Creation in the Canine Elbow—A Cadaveric Study

**DOI:** 10.3390/ani15040543

**Published:** 2025-02-13

**Authors:** Piotr Trębacz, Jan Frymus, Mateusz Pawlik, Anna Barteczko, Aleksandra Kurkowska, Joanna Berczyńska, Michał Czopowicz

**Affiliations:** 1Department of Surgery and Anesthesiology of Small Animals, Institute of Veterinary Medicine, Warsaw University of Life Sciences-SGGW, Nowoursynowska 159 C, 02-776 Warsaw, Poland; 2CABIOMEDE Ltd., Karola Olszewskiego 21, 25-663 Kielce, Poland; mateusz.pawlik@cabiomede.com (M.P.); anna.barteczko@cabiomede.com (A.B.); aleksandra.kurkowska@cabiomede.com (A.K.); 3Department of Biomaterials and Medical Devices Engineering, Faculty of Biomedical Engineering, Silesian University of Technology, Roosevelta 40, 41-800 Zabrze, Poland; 4Surgical Practice, Al. Niepodległości 24/30, 02-653 Warsaw, Poland; joanna.berczynska99@gmail.com; 5Division of Veterinary Epidemiology and Economics, Institute of Veterinary Medicine, Warsaw University of Life Sciences-SGGW, Nowoursynowska 159 C, 02-776 Warsaw, Poland; michal_czopowicz@sggw.edu.pl

**Keywords:** dog, arthroscopy, elbow, ulnar nerve

## Abstract

The elbow is the most commonly arthroscoped joint in dogs, and typically, a medial telescopic portal is used during this procedure. In contrast, some investigators prefer the caudo-medial arthroscopic portal, which allows better visualization of the medial and caudal elbow compartments. This requires the insertion of the arthroscope very close to the ulnar nerve. To date, only a few studies have reported the use of the caudo-medial portal, and no cases of ulnar nerve injury have been reported as a complication of this procedure. Therefore, we investigated the risk of ulnar nerve trauma following caudo-medial portal placement in canine cadavers. Consequently, after the telescope was inserted caudo-medially, a tissue incision was made to visualize the nerve, and the distance between the cannula and the nerve was measured. An injury was diagnosed when the telescope at least scratched the nerve. Nerve injury or high risk of trauma was observed in as many as 70% of the dogs. These findings suggest that injury of the ulnar nerve during the creation of the caudo-medial arthroscopic portal appears to occur quite often, which has not been considered before.

## 1. Introduction

The elbow is the most commonly arthroscoped joint in dogs. The medial approach is a traditional telescopic portal for this procedure [1,2]. It is located distally or caudo-distally to the tip of the medial epicondyle of the humerus. This portal is considered safe for the neurovascular structures surrounding the elbow and is often chosen [3,4]. Recently, some researchers have suggested that a caudo-medial arthroscopic portal allows better visualization of the medial and caudal elbow compartments, especially the medial humeral epicondyle [5,6]. However, this requires the insertion of the arthroscope in a very close vicinity to the ulnar nerve. This can be potentially harmful to the ulnar nerve. The ulnar nerve originates from the brachial plexus, receiving fibers from the ventral spinal roots of C7, C8, Th1, and Th2, with most coming from Th1 [7]. After leaving the brachial plexus, it accompanies the median nerve, and then in the distal part of the brachial region, it directs caudally to pass the elbow above the medial epicondyle of the humerus and provide sensory innervation to the skin of the caudal antebrachial region as well as the lateral part of the carpal, metacarpal, and digital area. It also provides motor innervation for two muscles—the flexor carpi ulnaris and the deep digital flexor. Therefore, ulnar nerve damage can result in hyperextension of the carpus during weight bearing [8,9]. No cases of ulnar nerve dysfunction have been reported as a complication of the caudo-medial arthroscopic approach; however, only in very few studies has this portal been used so far [5,6]. On the other hand, the above-mentioned flexors are predominantly innervated by the median nerve branches [10]. Hence, the motor activity of the forelimb and gait of the dog is usually only mildly affected by ulnar nerve dysfunction. This fact may render ulnar nerve injuries unnoticed without professional neurological examination, and the condition can remain underdiagnosed in routine veterinary practice. Therefore, the present study aimed to investigate the canine cadavers’ risk of ulnar nerve injury following caudo-medial arthroscopic portal placement.

## 2. Materials and Methods

The study included 30 dog cadavers euthanized for reasons unrelated to this study. According to Polish legislation, ethics approval was not required for this study (Act of the Polish Parliament of 15 January 2015 on the Protection of Animals Used for Scientific or Educational Purposes, Journal of Laws 2015, item 266) [11]. The cadavers were evaluated for signs of previous elbow surgery, deformity, extensive scarring, or contracture that could possibly alter the native course of the ulnar nerve. Each cadaver was placed in lateral recumbency with the limb arthroscoped downwards. The hair on the elbow joint was clipped, the skin was thoroughly cleaned, and the joint was filled with sterile 0.9% saline. Next, an arthroscopic sheath with a blunt obturator for the 1.9 mm telescope (Karl Storz, Tuttlingen, Germany) was inserted caudo-medially between the most prominent parts of the medial epicondyle of the humerus and the olecranon tuber of the ulna, as described by Danielski and Yeadon [5] (Figure 1). Then, with the arthroscopic sheath in place, a longitudinal incision of the skin and subcutaneous tissue was made along the palpable indentation line between the cranial border of the triceps muscle and the caudal aspect of the medial humeral epicondyle. After exposure of the ulnar nerve, the distance between the sheath and the ulnar nerve was measured using an electronic caliper (Engindot, Shenzhen, China) (Figure 2). The measurements were recorded in mm. The ulnar nerve injury was diagnosed when the arthroscopic sheath visibly penetrated or at least scratched the ulnar nerve (Figure 3). A high risk of ulnar nerve injury was defined if the distance between the arthroscopic sheath and the ulnar nerve was 0 mm, i.e., the cannula directly touched the nerve, but no visible damage was present (Figure 4).

## 3. Statistical Analysis

Numerical variables (age, body weight, and distances between the arthroscopic sheath and the ulnar nerve) were examined for normality using the normal probability Q–Q plots and the Shapiro–Wilk test. The normality assumption was held in the case of age (*p* = 0.306) and body weight (*p* = 0.831) but was violated in the case of distance (*p* < 0.001). Therefore, for the consistency of data presentation, all numerical variables were summarized as the median, interquartile range (IQR), and range and compared between unpaired groups (males vs. females; pedigree vs. crossbreeds; dogs with vs. dogs without ulnar nerve injury) using the Mann–Whitney U test. The distances between the ulnar nerve and the arthroscopic cannula were compared between elbows (paired groups) using the Wilcoxon signed rank test. Categorical variables (sex, breed, occurrence of ulnar nerve injury or high risk of ulnar nerve injury event) were expressed as the count and proportion and compared between elbows (paired groups) using McNemar’s test and between unpaired groups using the maximum likelihood G test exact test. The 95% confidence intervals (CI 95%) for proportions were calculated using the Wilson score method. All tests were two-sided. A significance level (α) was set at 0.05. Statistical analysis was performed in TIBCO Statistica 13.3 (TIBCO Software Inc., Palo Alto, CA, USA).

## 4. Results

In all cadavers, we did not find macroscopic pathologies of the elbows. The creation of the caudo-medial arthroscopic portal was successful in all 60 elbows. The study included cadavers of 15 males and 15 females, aged 6–14 years (median: 9 years, IQR: 8–11 years) and weighing 10–30 kg (median: 20 kg, IQR: 17–24 kg). Neither age (*p* = 0.674) nor body weight (*p* = 0.118) differed significantly between males and females. Fifteen dogs were crossbreeds, and the remaining fifteen pedigree dogs belonged to the eight following breeds: American Staffordshire Terrier and Border Collie (3 dogs of each breed), Bavarian Mountain Hound, Shetland Sheepdog and French Bulldog (2 dogs of each breed), Beagle, Springer Spaniel, and Collie. The distance between the arthroscopic cannula and the ulnar nerve ranged from 0 to 8.0 mm (median: 0.5 mm, IQR: 0–3.7 mm) in the left elbow and from 0 to 5.0 mm (median: 0.3 mm, IQR: 0–2.7 mm) in the right elbow and the distances did not differ significantly between sides (*p* = 0.297). The ulnar nerve injury occurred in 16/30 dogs (53%; CI 95%: 36%, 70%)—in 11 dogs, the injury was unilateral (7 dogs—left elbow, 4 dogs—right elbow) and, in 5 dogs, it was bilateral. The ulnar nerve injury affected twelve left and nine right elbows, and the difference in the ulnar nerve injury occurrence between the left and right elbow was not significant (*p* = 0.547). A high risk of an ulnar nerve injury event was observed in eight dogs—five dogs without ulnar nerve injury in the opposite joint and three dogs with contralateral ulnar nerve injury.

In total, ulnar nerve injury or high risk of injury events were observed in 21/30 dogs (70%; CI 95%: 52%, 83%)—they were unilateral in 12 dogs (6 left and 6 right elbows) and bilateral in 9 dogs, so in total, they affected 15 left and 15 right elbows. The occurrence of ulnar nerve injury or high risk of injury events was not significantly associated with sex (*p* = 0.427), breed (*p* = 0.109), age (*p* = 0.233), or body weight (*p* = 0.666) (Table 1).

## 5. Discussion

Postoperative ulnar neuropathy is an injury manifesting in the sensory or motor distribution of the ulnar nerve after anesthesia or surgery. In human medicine, ulnar nerve injury has been reported as a direct result of orthopedic surgery, such as repair of supracondylar humerus fractures, distal humerus fractures, ulnar nerve transposition surgery, and elbow arthroscopy during the creation of the posteromedial portal for the telescope [12,13]. Hilgersom et al. reported that, in humans, the most frequently injured nerve in the elbow during arthroscopy is the ulnar nerve (38–42%), with other nerves at risk being the superficial radial (22–33%), posterior interosseous (8–19%), median (0–10%), anterior interosseous (5–8%), and medial (5–8%), lateral, and posterior antebrachial cutaneous nerves. The injury to the ulnar nerve at the level of the medial humeral epicondyle usually causes sensory abnormalities in the fourth and fifth fingers, motor weakness in finger flexion, and finger abduction of four and five fingers [13]. A common sign of severe ulnar nerve injury is the ’ulnar claw’ hand [12,14]. To increase the safety of the arthroscopic approach, it is recommended to draw anatomical landmarks, identify and protect the ulnar nerve, and create neurolysis in the event of ulnar nerve compression or elbow stiffness [15,16]. Elbow flexion and joint distention are additional procedures to increase the safety of elbow arthroscopy, as they increase the nerve-to-portal distance [13]. In another study, Hilgersom et al. also recommended that ulnar nerve palpation be standard practice before every elbow arthroscopy [15].

We found no guidelines for ulnar nerve protection in the veterinary literature. Jardel et al. [3] and McCarthy [4] suggested palpation of the ulnar nerve before canine elbow arthroscopy and maintaining joint position while puncturing the joint. Jardel et al. [3] reported that in 11 cadavers of large-breed dogs (20 elbows), after creating the traditional medial portal for the telescope, the distance between the arthroscopic sheath and the ulnar nerve was between 1 and 7 mm. They did not notice injury to the nerve and found this portal safe. However, they also suggested that the caudo-medial portal could harm the ulnar nerve. This approach can be more valuable than the traditional one for a more accurate inspection of the medial and caudal elbow compartment. Using this portal, Danielski and Yeadon [5] reported a previously unreported cartilage lesion on the caudal aspect of the medial humeral condyle identified in dogs with humeral intracondylar fissures. They examined 21 elbows in 14 spaniel-breed dogs. They noticed that the caudo-medial portal was not associated with apparent iatrogenic cartilage damage to the weight-bearing articular surface of the medial humeral condyle and ulna, even during joint manipulation. The authors did not consider the ulnar nerve injury. At the same time, the authors suggested that it was not known whether the learning curve for this caudo-medial arthroscopic portal might differ from that of the conventional medial arthroscopic portal. Similarly, they speculated if iatrogenic damage of the cartilages and/or to the periarticular structures, such as the ulnar nerve, might be more likely when performed by less experienced arthroscopists, particularly when combined with intraoperative joint manipulation. Ultimately, the authors suggested this should be an area for further investigation. In another study, Danielski et al. used the caudo-medial arthroscopic portal for elbow arthroscopy in 35 spaniel-breed dogs (51 elbows). In this study, the authors did not consider ulnar nerve injuries [6].

The main finding of our study is that the caudo-medial arthroscopic portal was associated with direct ulnar nerve injury in roughly 50% of dogs and posed a threat of nerve injury in approximately 20% of animals. Given a relatively non-numerous study population, these observations should be treated cautiously and warrant further verification in clinical conditions. However, our results potentially indicate that this arthroscopic portal may be unsafe for the ulnar nerve. We have reached a high rate of ulnar nerve injuries in our study compared to the studies mentioned above. The medial telescopic portal originally described by Van Ryssen et al. [1] is routinely used in our institution in clinical practice, and no complications result. This portal is considered safe for the neurovascular structures surrounding the elbow. The relatively large distance between this portal and the ulnar nerve makes it difficult to damage the nerve. The caudo-medial portal, which is close to the ulnar nerve, may be more prone to nerve damage. A second possible reason for this high incidence of ulnar nerve injury may be a withdrawal of identification of the course of the ulnar nerve before creating the caudo-medial portal. The discrepancy between our results and those reported in clinical studies is noteworthy. The reason could be the poor clinical manifestations of post-arthroscopic ulnar nerve injuries, which led to underdiagnosis in the aforementioned clinical studies. The ulnar nerve below the elbow gives muscular branches to the carpal ulnar flexor, deep and superficial digital flexors, and the interosseus muscle [7,8,17]. The skin over the proximal part of the antebrachium from the medial to the caudolateral aspects of the distal part of the antebrachium and the proximal two-thirds of the skin of the caudolateral aspect of the antebrachium is also innervated by the ulnar nerve, along with other nerves. The autonomous zone for the ulnar nerve is the skin over the lateral aspect of the fifth digit [13,18]. Basa et al. [9] have surgically treated dogs with neurofibroma of the ulnar nerve in the carpal canal. The signs of sensory or motor deficits were not detected in this case. The authors also overlooked the hyperextension of the carpus. They speculated that despite the ulnar nerve injury being reported to cause hyperextension of the carpus when weight-bearing on the limb, in this case, there were no detectable abnormalities on follow-up examination because the radial nerve (innervates extensors of the thoracic limb) was intact. So, it should be noted that dogs’ ulnar nerve dysfunction can be challenging to identify.

As our study was performed on cadavers, we could not verify to what extent a direct ulnar nerve injury or high risk of injury events associated with caudo-medial arthroscopic portal creation affected the motor and sensory function of the ulnar nerve. However, the discouraging results of our study suggest that identification of the ulnar nerve should be a mandatory procedure before creating the caudo-medial portal. The nerve can be palpated as it crosses the medial epicondyle [4,9].

## 6. Conclusions

Ulnar nerve injury during caudo-medial arthroscopic portal creation appears more common. This approach should only be used if the course of the ulnar nerve in the medial humeral epicondyle region can be precisely determined.

## Figures and Tables

**Figure 1 animals-15-00543-f001:**
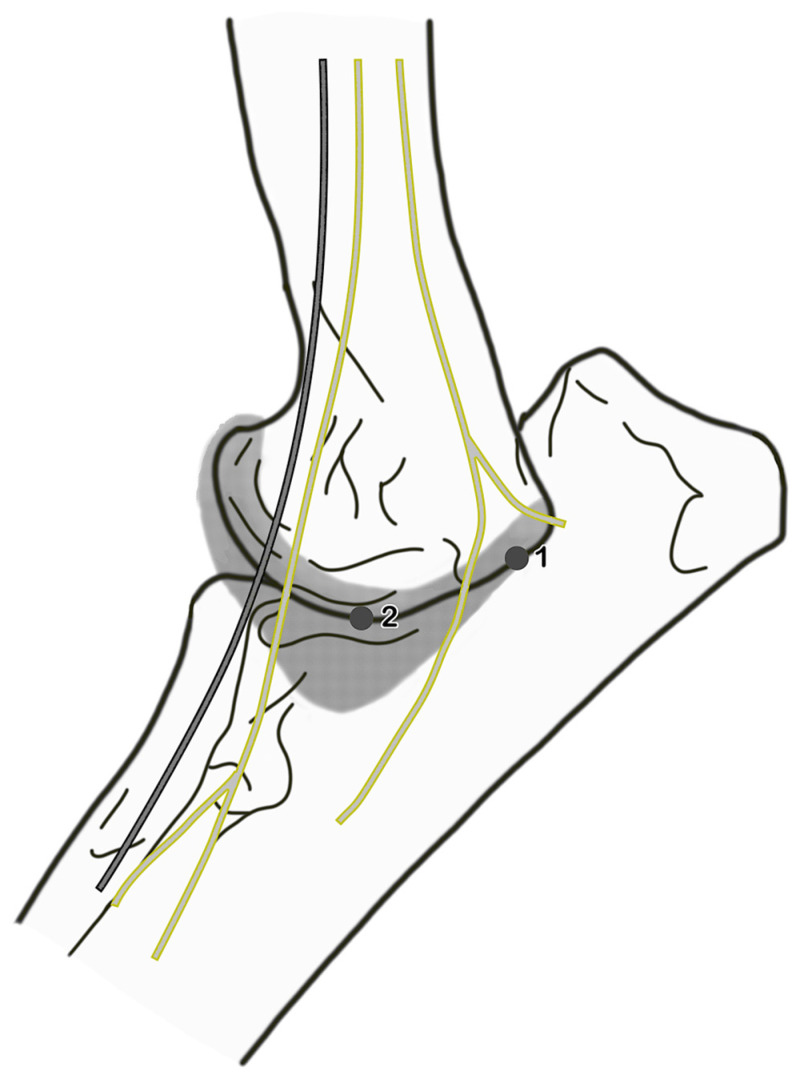
Medial view of the elbow: portal locations for arthroscopy; 1, caudo-medial arthroscopic portal near the ulnar nerve; 2, egress portal.

**Figure 2 animals-15-00543-f002:**
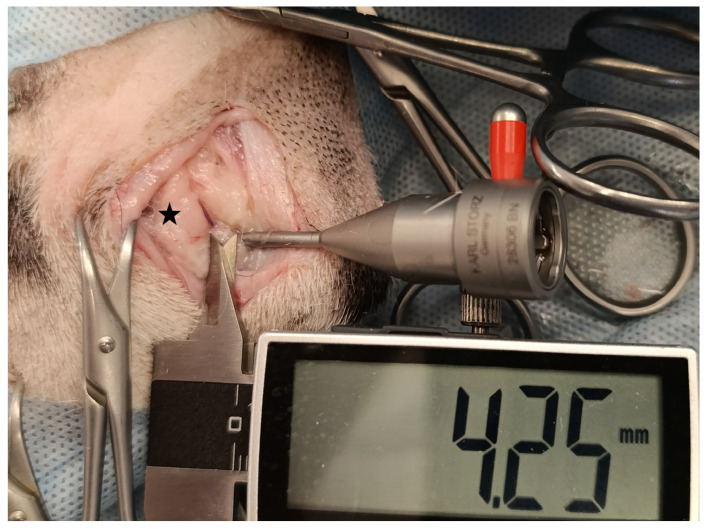
The distance between the ulnar nerve (black star) and the arthroscopic sheath was measured in mm by the electronic caliper—the right elbow of a male Bavarian Mountain Dog.

**Figure 3 animals-15-00543-f003:**
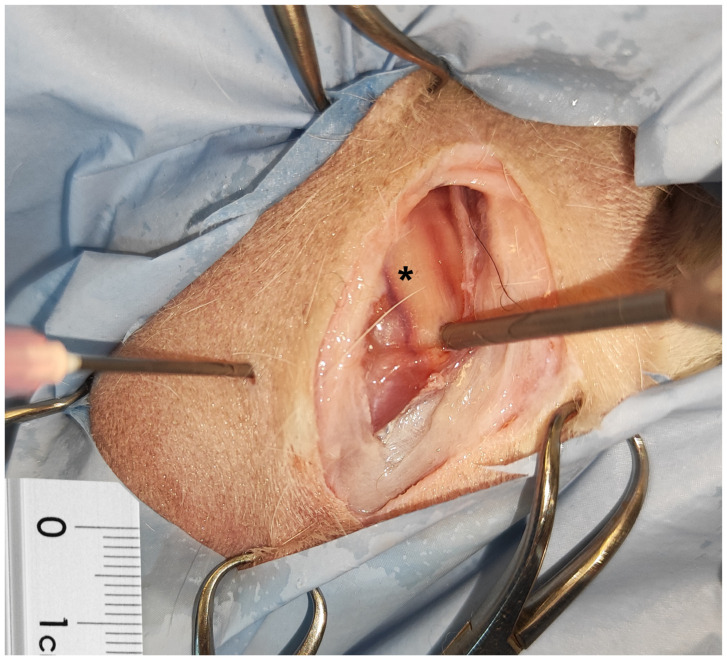
Direct puncture of the ulnar nerve (black star) after inserting the arthroscopic sheath in the right elbow of a male border Collie.

**Figure 4 animals-15-00543-f004:**
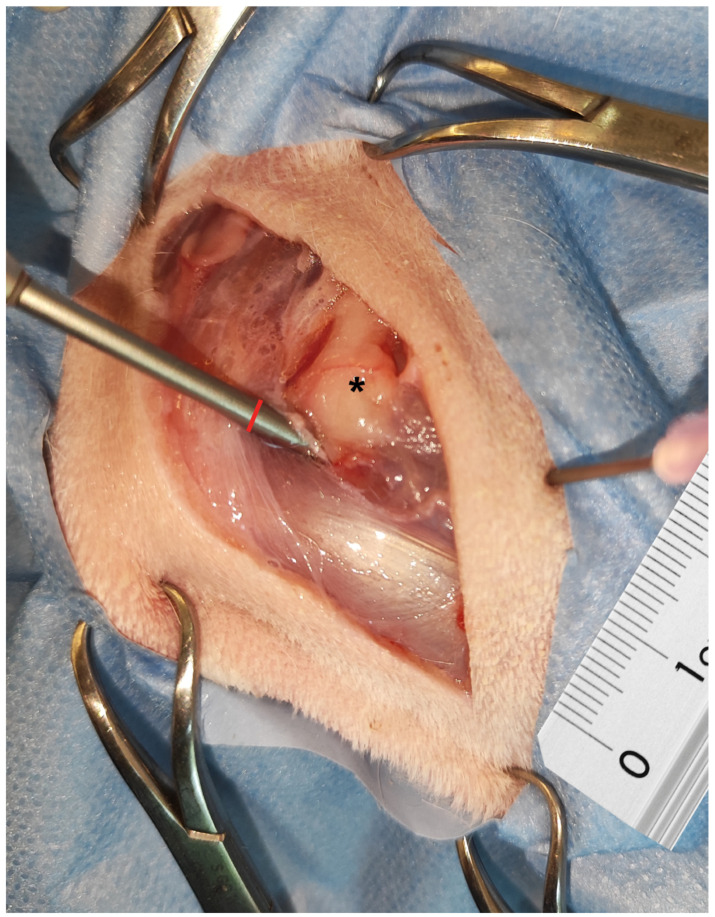
High risk of ulnar nerve injury event. The distance between the arthroscopic sheath and the ulnar nerve (black star) is 0 mm—the left elbow of a male border Collie.

**Table 1 animals-15-00543-t001:** The association between the occurrence of ulnar nerve injury or high risk of injury event and demographic characteristics of dogs.

Demographic Characteristics	Ulnar Nerve Injury or High Risk of Injury Event		*p*-Value
	Present (*n* = 21)	Absent (*n* = 9)	
Male sex	12 (57%)	3 (33%)	0.427
Pedigree status	8 (38%)	7 (78%)	0.109
Age [years]	9, 8–10 (7–12)	11, 9–11 (6–14)	0.233
Body weight [kg]	20, 16–23 (10–29)	20, 17–24 (14–30)	0.666

## Data Availability

The data presented in this study are available in the article.

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
