# Peer review of "Risk of Ulnar Nerve Injury Following Caudo-Medial Arthroscopic Portal Creation in the Canine Elbow—A Cadaveric Study"

_animals, 2025, doi:10.3390/ani15040543_

Round 1
Reviewer 1 Report
Comments and Suggestions for Authors
Line 16: arthroscoped
Line 21/22: Grammar: Please change to: Therefore, we investigated the risk of ulnar nerve trauma following caudo-medial portal placement in canine cadavers.
Line 45: arthroscoped
Line 84: reference missing
Line 85: spacing between words
Line 132: Collie
Author Response
Dear Reviewer,
Thank you for your valuable feedback and comments.
Line 16: arthroscoped
answer: We have corrected it
Line 21/22: Grammar: Please change to: Therefore, we investigated the risk of ulnar nerve trauma following caudo-medial portal placement in canine cadavers.
answer: We have corrected it
Line 45: arthroscoped
answer: Line 52 We have corrected it
Line 84: reference missing
answer: We add
Line 85: spacing between words
answer: We have corrected it
Line 132: Collie
answer: We have corrected it

Reviewer 2 Report
Comments and Suggestions for Authors
This is an interesting paper investigating the incidence of ulnar nerve injury following the caudo-medial portal placement using cadavers. Although the technique is not commonly used, this information should be helpful for surgeons.
Overall, this paper is well written, and my only concern is below.
Line 179 and 188 – the 2 papers the authors cited did not describe any sentence like “no ulnar nerve injuries were detected”; the authors most likely did not actually check if there was a nerve injury from scope during the surgery, so some dogs might have nerve injury without symptoms. Therefore, the word “reported” misleads readers, please change.
Some minor comments are below.
Line 84- The reference number is missing
Line 135-137 – Was there any tendency to which part of the ulnar nerve was injured? (e.g. caudal margin of the ulnar nerve (close to the olecranon) was injured in XXX% of cases)
Table1. Please describe what each number shows in age and body weight. (mean, median, SD, SE, range, IQR…)
Author Response
Dear Reviewer,
Thank you for your valuable feedback and comments. We have made the necessary revisions. We believe these changes will significantly enhance the quality and value of the paper.
Line 179 and 188 – the 2 papers the authors cited did not describe any sentence like “no ulnar nerve injuries were detected”; the authors most likely did not actually check if there was a nerve injury from scope during the surgery, so some dogs might have nerve injury without symptoms. Therefore, the word “reported” misleads readers, please change.
answer: We have corrected it
Line 210-220: The authors did not consider the ulnar nerve injury. At the same time, the authors suggested that it was not known whether the learning curve for this caudo-medial arthroscopic portal might differ from that of the conventional medial arthroscopic portal. Similarly, they speculated if iatrogenic damage of the cartilages and/or to the periarticular structures, such as the ulnar nerve, might be more likely when performed by less experienced arthroscopists, particularly when combined with intraoperative joint manipulation. Ultimately, the authors suggested this should be an area for further investigation. In another study, Danielski et al. used the caudo-medial arthroscopic portal for elbow arthroscopy in 35 spaniel-breed dogs (51 elbows). In this study, the authors did not consider the ulnar nerve injuries (4).
Line 84- The reference number is missing
answer: We have corrected it
Line 135-137 – Was there any tendency to which part of the ulnar nerve was injured? (e.g. caudal margin of the ulnar nerve (close to the olecranon) was injured in XXX% of cases)
Answer: Yes, but because the presence of nerve injury is a categorical variable, we use the word “association,” not “correlation,” as correlation, by definition, applies to two numerical or at least ordinal variables.
Table1. Please describe what each number shows in age and body weight. (mean, median, SD, SE, range, IQR…)
Answer: Statistical measures used in the table 1 have been described.

Reviewer 3 Report
Comments and Suggestions for Authors
This article provides important information about the risk of ulnar nerve injury during medial arthroscopic entrance surgery at the tail of the elbow in dogs, and recommends taking extra precautions to protect the ulnar nerve when performing such surgery. This manuscript is interesting, but there are still several issues that need to be addressed:
1. Please discuss whether pain and loss of flexion function would affect the results.
2. Please discuss if there is delayed ulnar neuropathy.
3. Please explain why this portal was used in very few studies in line 61.
4. Please list the reasons why the study has such a high injury rate in line 194.
5. Please add the scale bar.
6. An anatomical landmark diagram can be added detailing the measurement method of the caudal medial arthroscopic portal.
7. The discussion section should analyze the results of the study in depth.
8. Please cite all the closely relative references, especially those directly related to the study design, methods, and interpretation of results.
9. The article mentions that under Polish law, this study does not require ethical approval. Authors are advised to provide more information on why ethical approval is not required for a particular type of study and whether this is in line with international standards.
10. The correlation between dogs of different body sizes and the occurrence of ulnar nerve injury or high-risk injury events is described in line 146.
Author Response
Dear Reviewer,
Thank you for your valuable comments. We have made the necessary revisions, which we believe will significantly enhance the quality and value of this paper.
This article provides important information about the risk of ulnar nerve injury during medial arthroscopic entrance surgery at the tail of the elbow in dogs and recommends taking extra precautions to protect the ulnar nerve when performing such surgery. This manuscript is interesting, but there are still several issues that need to be addressed:
- Please discuss whether pain and loss of flexion function would affect the results.
Answer: The identification of ulnar nerve dysfunction is difficult. In one case report, the dog did not demonstrate signs of sensory or motor front limb dysfunction even after the excision of the tumor of the ulnar nerve. As our study was performed on cadavers, we could not verify to what extent a direct ulnar nerve injury or high risk of injury event associated with caudo-medial arthroscopic portal creation affected the motor and sensory function of the ulnar nerve. We have added the above case to the discussion section.
Line 174-188: Postoperative ulnar neuropathy is an injury manifesting in the sensory or motor distribution of the ulnar nerve after anesthesia or surgery. In human medicine, ulnar nerve injury has been reported as a direct result of orthopedic surgery, such as repair of supracondylar humerus fractures, distal humerus fractures, ulnar nerve transposition surgery, and elbow arthroscopy during the creation of the postero-medial portal for the telescope (9, 10,). Hilgersom et al. (10) reported that in humans the most frequently injured nerve in the elbow during arthroscopy is the ulnar nerve (38%-42%), with other nerves at risk being the superficial radial (22%-33%), posterior interosseous (8%-19%), median (0%-10%), anterior interosseous (5%-8%), and medial (5%-8%), lateral, and posterior antebrachial cutaneous nerves. The injury to the ulnar nerve at the level of the medial humeral epicondyle usually causes sensory abnormalities in the fourth and fifth fingers, motor weakness in finger flexion, and finger abduction of 4 and 5 fingers.
- Please discuss if there is delayed ulnar neuropathy.
Answer: there is a lack of information about delayed ulnar neuropathy in the veterinary literature. As our study was performed on cadavers, we could not verify to what extent an ulnar nerve injury associated with caudo-medial arthroscopic portal creation affected the motor and sensory function of the ulnar nerve.
- Please explain why this portal was used in very few studies in line 61.
Answer: As you suggested, we have rewritten this part of the paper and added more information: “ The elbow is the most commonly arthroscopied joint in dogs. The medial approach is a traditional telescopic portal for this procedure. It is located distally or caudo-distally to the tip of the medial epicondyle of the humerus. This portal is considered safe for the neurovascular structures surrounding the elbow and is often chosen. Recently, some researchers have suggested that a caudo-medial arthroscopic portal allows better visualization of the medial and caudal elbow compartments, especially the medial humeral epicondyle (3,4). However, this requires the insertion of the arthroscope in the very close vicinity of the ulnar nerve. It can be potentially harmful to the ulnar nerve. “
- Please list the reasons why the study has such a high injury rate in line 194.
Answer: We listed the proximity between the portal and the ulnar nerve, the lack of identification of the ulnar nerve’s course before creating the portal and discussed the discrepancy between our results and those reported in clinical studies.
- Please add the scale bar.
answer: We add Figure 3 and 4
- An anatomical landmark diagram can be added detailing the measurement method of the caudal medial arthroscopic portal.
answer: We add Figure 1 line 109
- The discussion section should analyze the results of the study in depth.
Answer: We added a paragraph about the possible reasons for the discrepancy between our results and those reported in clinical studies, and we have detailed information on the innervation of the antebrachium and digits by the ulnar nerve.
line 231-255
- Please cite all the closely relative references, especially those directly related to the study design, methods, and interpretation of results.
answer: We added citations
- The article mentions that under Polish law, this study does not require ethical approval. Authors are advised to provide more information on why ethical approval is not required for a particular type of study and whether this is in line with international standards.
Answer: Following the European lawmaker, the Polish legislator indicates the acceptable uses of animals. These uses comprise research, ensuring animal welfare or improving livestock rearing or breeding conditions, developing and manufacturing therapeutic products, protecting the natural environment, and academic education or training. The lawmaker defines animal use for the abovementioned purposes as a procedure. More accurately, a method is “any form of use of animals for the purposes which may cause the animal a level of pain, suffering, distress or lasting harm, to the extent equivalent to, or greater than that caused by the introduction of a needle. In contrast, animal carcasses are category 1 waste, are not protected, and must be disposed of in a controlled manner.
We have corrected it we included the source (citation) line 86-88
- The correlation between dogs of different body sizes and the occurrence of ulnar nerve injury or high-risk injury events is described in line 146.
Answer: Yes, but because the presence of nerve injury is a categorical variable, we use the word “association,” not “correlation,” as correlation, by definition, applies to two numerical or at least ordinal variables.

Round 2
Reviewer 3 Report
Comments and Suggestions for Authors
the revised manuscript has been well improved to meet all the requirements.